# Multiomics Studies on the Effects of High-Temperature Stress on Male Sterility in *Gossypium barbadense*

**DOI:** 10.3390/ijms26083693

**Published:** 2025-04-14

**Authors:** Jiangbo Li, Xiaojuan Deng, Man Gao, Tao Lv, Yongsheng Cai, Yanying Qu, Quanjia Chen, Kai Zheng

**Affiliations:** Xinjiang Key Laboratory of Crop Biology Breeding, College of Agriculture, Xinjiang Agricultural University, Urumqi 830052, China; 18699160581@163.com (J.L.); dengxj007@163.com (X.D.); 13209972635@163.com (M.G.); 13909447464@163.com (T.L.); cys0620@126.com (Y.C.); xjyyq5322@126.com (Y.Q.); chqjia@126.com (Q.C.)

**Keywords:** *Gossypium barbadense*, transcriptome, metabolome, anther, male sterility

## Abstract

High-temperature (HT) stress has been recognized as one of the main factors restricting the normal growth and development of cotton and severely affects fiber quality and yield. To elucidate the regulatory mechanism of male sterility-related hormones in *Gossypium barbadense* under HT stress, we explored candidate genes closely related to male sterility in *G. barbadense*. We studied the expression profiles of hormones and genes in the anthers of *G. barbadense* GB150 under HT stress by combining transcriptomic and metabolomic analyses. Through a combined analysis of the transcriptional metabolism of the anthers of *G. barbadense* GB150, we determined the contents of ABA, JA, SA, IAA, tZR, and GA_20_ and the expression of genes related to biosynthetic pathways and signal transduction pathways. The results revealed that the ABA and JA contents significantly increased after HT; the IAA, tZR, and GA_20_ contents significantly decreased; and the SA content did not significantly change after HT. We then used weighted gene coexpression network analysis (WGCNA) to further analyze the interactions among hormones, transcription factors, and core genes and constructed hormone coexpression networks and genome-wide coexpression networks. Through these network analyses, we ultimately identified 10 candidate genes closely related to male sterility in *G. barbadense*. Using qRT-PCR, resequencing data from 221 *G. barbadense* materials revealed that ALA4 (*Arabidopsis thaliana* has been proven to be associated with male fertility) and SBP1 (two stop gains in the gene structure) may play important roles in the process of male sterility in *G. barbadense*. The results of this study provide a theoretical basis for the molecular mechanism of male sterility in *G. barbadense*.

## 1. Introduction

High-temperature (HT) stress is defined as irreversible damage to the growth and development of plants caused by a temperature exceeding a certain threshold level within a certain period. When the temperature increases frequently to 10–15 °C above the ambient temperature in a short period, it is called HT shock or HT stress [1]. Cotton is an annual or perennial herb of the *Malvaceae* family, and its optimum growth temperature is 20–30 °C [2]. *Gossypium barbadense* is very sensitive to HT stress, and HT stress inhibits anther dehiscence, affecting pollen vitality and resulting in boll fall and reduced yield. Moreover, continuous HT stress affects the development of fiber quality, resulting in a decrease in the fiber strength and fiber uniformity [3]. In addition, HT stress can affect plant reproductive organs and vegetative organs, but HT has a greater effect on reproductive organs than on vegetative organs [4].

The decrease in the cotton yield is due mainly to HT, which increases the possibility of male sterility [5,6]. Previous studies have shown that HT affects several key stages of cotton reproductive growth [7]. During the reproductive organ development stage, HT causes the pollen quantity and stamen vitality to decrease significantly, resulting in the formation of deformed pollen grains, weakening pollination function, and eventually leading to male sterility and abortion. Moreover, the pistil is also negatively affected by HT: the female gametophyte is degraded, the style and ovary are deformed, and the structural integrity of the pistil and the function of the ovule are reduced. In the process of pollination to fertilization, HT also has adverse effects, such as reduced pollen adhesion and germination, hindered pollen tube growth, and reduced ovule viability and lifespan, which ultimately affect normal gamete union.

Under HT stress, the above factors hinder improvements in cotton yield. If no mitigation measures are taken as the climate warms, the global average surface temperature will increase (2.6–4.8 °C) by the end of the century, with every 1 °C increase in temperature affecting crop yields by up to 17% during the growing season [8]. HT stress has become an important limiting factor affecting cotton production. To reduce the impact of HT on the cotton industry, it is necessary to accelerate the collection and identification of HT-resistant cotton resources, analyze the molecular mechanism of HT resistance, and create new HT-resistant materials.

Combined transcriptomic and metabolomic analysis is conducive to obtaining a more comprehensive understanding of biological processes and their regulatory networks inside plants and is widely used in various organisms [9,10,11,12,13,14]. Joint analyses can not only identify novel metabolic pathways and key metabolites, which is highly important for the study of secondary plant metabolism and the production of chemical substances, but also break traditional boundaries and promote crossover and fusion between different research fields [15].

Under HT stress, dynamic changes in hormones directly affect the normal growth and development of plants [16]. Abscisic acid (ABA), an important hormone that regulates the water state of plants, plays an important role in HT stress. By regulating transpiration and water absorption, ABA can meet the demand of atmospheric CO_2_ for photosynthesis [17]. *Arabidopsis* mutants that are not sensitive to ABA present significant defects in HT resistance [18]. As a stress-related hormone, abnormal jasmonic acid content also leads to pollen inactivation and anther dehiscence. Auxin, gibberellin, and cytokinin, key hormones in plant growth and development, have also been confirmed to play roles in male plant sterility [19].

The aim of this study was to explore the regulatory mechanism by which HT stress affects medicinal anther dehiscence in *G. barbadense*. First, 36 cotton materials were selected from 534 *G. barbadense* resources according to the extent of four flower traits (style length, stigma length, single anther length, and stamen length). Among these materials, the GB150 material was selected as a key research object because of its long anther length. To assess the heat resistance of these materials, 36 resources were identified at the seedling and adult stages. The results revealed that GB150 was highly sensitive to HT stress during different periods (Appendix A). Moreover, in the field warming stress experiment, we observed that the GB150 material presented a nondehiscent anther phenotype, whereas the other materials did not. To date, many studies have investigated the molecular mechanism of male sterility in cotton [20,21], but the relationship between male sterility and hormone-related regulatory networks in *G. barbadense* under HT stress is still relatively uncharacterized. To analyze the response mechanism of *G. barbadense* to male sterility and HT stress, it is important to study the regulatory network of hormone metabolism and signal transduction. Therefore, we conducted in-depth hormone and transcriptomic analyses of GB150 to analyze the interaction between male sterility and hormone metabolism in *G. barbadense* under HT stress. Based on a detailed analysis of the interactions between hormones and hormone signaling pathways, we constructed a transcriptional regulation model of male sterility in *G. barbadense* under HT stress and identified 10 related candidate genes. These findings provide a foundation for further analysis of the molecular mechanism of male sterility in *G. barbadense*.

## 2. Results and Analysis

### 2.1. Quality Assessment of the Transcriptomic and Metabolomic Sequencing Data

We constructed an RNA-seq library using six anther samples of the *G. barbadense* resource GB150 under HT stress and normal conditions. A total of 61.74 Gb of clean data was obtained, and more than 6.43 Gb of clean data was obtained for each sample. The total reads of each sample were subsequently compared with the reference genome of *G. barbadense*. The efficiency of the comparison was 93.82–95.87%, the percentage of Q30 bases was no less than 94.28%, and the GC content was no less than 43.76% (Table 1). These results confirmed that the RNA-seq data in this study were of high quality and could be used for subsequent analysis.

We conducted a principal component analysis of the expressed genes (Figure 1A) and found that samples in the same group were clustered into one class, and samples in different groups with anther dehiscence (CK group) or anther nondehiscence (HOT group) were far apart, indicating that the expression levels of anther-related genes under the two treatments greatly differed. The principal component results indirectly demonstrated the reliability of the transcriptomic data.

The abundances of anther-related metabolites of GB150 subjected to different treatments were determined, and the differences in metabolite abundances among the different treatment groups were analyzed. The metabolite PCA under different treatments revealed that the contribution rate of PC1 was 80.38%, and the contribution rate of PC2 was 7.51% (Figure 1B). The data in the two groups had good reproducibility, and there were significant differences in metabolite abundances between the groups.

### 2.2. Effects of High-Temperature Stress on Floral Organ Development and Hormonal Regulation in G. barbadense GB150

Many studies have shown that HT stress has significant effects on many aspects of plant morphology, photosynthesis, and the antioxidant system. In this study, we found that GB150 resources caused no anther dehiscence and decreased pollen viability following HT stress (Figure 2A,B). To explore the effects of HT stress on the reproductive development of *G. barbadense*, we investigated the relationships among the four flower traits of GB150 (Figure 2C). The results revealed a significant correlation between the stylar length and stigma length among the four flower traits but no significant correlation between the other flower traits. We found that of the four flower traits, only the stigma length was greater than that of the normal control group under HT stress, and the values of the other three flower traits (style length, stamen length, and single anther length) were shorter than those of the control group following HT stress. We systematically analyzed the correlations between the levels of six hormones and four flower traits and found that the levels of tZR, IAA, and GA_20_, three hormones that are involved mainly in plant growth and development, were significantly correlated with four flower traits, indicating that these hormones may play important roles in plant growth and development, especially flower formation and development. The tZR levels were significantly correlated with two indicators (style length and stamen length), the IAA levels were significantly correlated with two indicators (style length and stamen length), and the GA_20_ levels were significantly correlated with one indicator (style length). The levels of the hormones ABA, JA, and SA, which are involved in the plant stress response and defense response, were not significantly correlated with the four flower traits, which does not mean that they have no role in plant growth and development; they may not directly participate in or affect the development of flower traits.

### 2.3. Changes in ABA Content and Expression Levels in G. barbadense Anthers Under HT Stress

Under HT stress, changes in the ABA content in plants play a core role. Through in-depth studies of the biosynthesis and catabolic pathways of ABA in many plants, we elucidated its regulatory mechanism. In this study, the ABA contents of anthers subjected to HT stress were determined (Figure 3A). Compared with normal conditions, HT stress significantly increased the endogenous ABA content in anthers (Figure 3B). For the RNAseq expression analysis, we examined the expression patterns of differentially expressed genes (DEGs) involved in the ABA biosynthesis pathway (Figure 3C,D). Among these genes, VDE1 (Gbar_A02G014800, Gbar_A11G025550) and NCED5 (Gbar_A06G014870) presented upregulated expression, and the ABA2 (1), NCED1 (2), CYP707A1 (1), and CYP707A2 (2) genes presented downregulated expression. By analyzing ABA signal transduction pathway-related DEGs, we found that ABF2 (Gbar_A05G025600) and ABF3 (Gbar_A12G002250) expression levels were upregulated and that ABF2 (Gbar_D05G020810) expression was downregulated.

qRT-PCR was performed on eight core genes (10) of the ABA biosynthesis and signal transduction pathways (Figure 3C). The expression levels of ABF2 (Gbar_A05G025600), VDE1 (Gbar_A11G025550), and ABF3 (Gbar_A12G002250) were upregulated, and the expression levels of the other seven genes were downregulated, consistent with the RNA-seq results. Further analysis revealed that 5 of the 14 ABA-related DEGs were upregulated and that 9 were downregulated. Based on these results, we speculated that the core genes closely related to the ABA content in anthers mainly negatively regulate ABA biosynthesis and signal transduction pathways under HT stress.

### 2.4. Effects of HT Stress on JA and SA Biosynthesis and Signal Transduction

JA and SA, as regulators of HT stress, also play important roles in the response to HT. The results revealed that the JA content significantly increased under HT stress (Figure 4A). Among the four types of genes encoding JA biosynthesis-related proteins (AOS3 (one), LOX5 (one), OPR3 (one), and CYP94B3 (three), for a total of six genes) (Figure 4C), the expression levels of two genes, LOX5 (Gbar_D09G022770) and OPR3 (Gbar_A06G007940), were upregulated, and the expression levels of the other four genes were downregulated. However, among the two classes of genes in the JA signal transduction pathway (MYC2 (one) and JAR1 (three), for a total of four genes), only one gene presented upregulated expression (MYC2 (Gbar_D09G020720)), whereas the other three genes presented downregulated expression (Figure 4E).

However, under HT stress, the SA content decreased, but the change was not obvious (Figure 4B). Therefore, we thoroughly analyzed the expression levels of core genes in the SA biosynthetic and signaling pathways. In the biosynthetic pathway (Figure 4D), the expression levels of three types of genes (for a total of six genes), ICS2 (three), UGT74F2 (two) and BSMT1 (one), tended to be downregulated. The changes in expression were consistent with the decrease in SA content, indicating that these SA biosynthesis-related genes expressed in anthers positively regulate SA synthesis after HT stress. In the signal transduction pathway, we observed expression changes in three types of core genes (twelve genes in total): TGA (six), NPR (one), and PAL (five). The results revealed six genes with upregulated expression and six genes with downregulated expression, among which five TGA genes (of six) presented upregulated expression, and only one TGA gene (Gbar_A12G009960) was downregulated (Figure 4F). Notably, the expression levels of SA-related core genes did change, but the SA content did not fluctuate significantly. Based on these findings, we hypothesized that although SA is involved in anther regulation after HT stress, it is not the main regulatory hormone.

### 2.5. Effects of HT Stress on IAA, GA_20_ and tZR Biosynthesis and Signal Transduction

The change in the IAA content under HT stress was very complicated. We found that the IAA content decreased significantly after HT stress (Figure 5A). However, the expression levels of both the CYP83B1 genes (Gbar_A10G020760 and Gbar_D10G021010) involved in the IAA biosynthesis pathway (Figure 5B) were upregulated. Moreover, among the four types of core genes (47 in total) in the signaling pathway, ARF (1), SAUR (21), AUX (3), and IAA (22), 12 genes (ARF (1), IAA (5), and SAUR (6)) presented upregulated expression, and the expression levels of 35 genes (AUX (3), IAA (17), SAUR (15)) were downregulated (Figure 5C). These results suggest that the expression levels of most signal transduction-related genes in anthers are downregulated under HT stress, which may lead to the inhibition of IAA expression and thus affect anther dehiscence.

The tZR content decreased significantly under HT stress (Figure 5D). To explore the molecular mechanisms underlying this phenomenon, we analyzed the expression patterns of genes involved in tZR biosynthesis (Figure 5E) and signal transduction. In the tZR biosynthesis pathway, we focused on four classes of genes (18 in total), including AHP (5), miaA (1), ZOG (2), and CKX (10). The expression levels of 6 genes (AHP (1), ZOG (1), and CKX (4)) were increased, whereas the expression levels of the other 12 genes (AHP (4), miaA (1), ZOG (1), and CKX (6)) were decreased. Moreover, in the tZR signal transduction pathway, we focused on the expression levels of ARR genes (5). The results revealed that the expression levels of four ARR genes were decreased, whereas that of only one ARR gene (Gbar_A09G025240) was increased (Figure 5F). Therefore, we concluded that the significant reduction in the tZR content under HT stress may be closely related to changes in the expression patterns of genes related to biosynthesis and signal transduction.

Under HT stress, the GA_20_ content also tended to decrease (Figure 5G). GA2OX plays a key role in the biosynthesis of GA_20_ (Figure 5H). However, we observed that only one GA2OX-related gene (Gbar_A01G002810) was increased under HT stress, whereas the expression levels of the remaining four GA2OX genes were decreased. We also focused on nine genes involved in the GA_20_ signaling pathway, including DELLA (two), GID (five), and PIF (two). Among these genes, the expression levels of five genes (GID (three) and PIF (two)) were increased, whereas the expression levels of the other four genes (DELLA (two), GID (two)) were decreased (Figure 5I). In summary, the decrease in the GA_20_ content under HT stress may be related to the downregulation of the expression of the GA20 biosynthesis-related gene GA2OX and the complex regulation of the related genes DELLA, GID, and PIF in the signal transduction pathway.

### 2.6. Changes in the Whole Transcriptome Under HT Stress

We conducted an in-depth analysis of the transcriptomic data, and through comparative analysis, we identified 14,857 DEGs (Figure 6A) that significantly differed between the two conditions. Among these DEGs, 5131 genes presented upregulated expression, and 9726 genes presented downregulated expression (Figure 6B). Through a hierarchical cluster analysis, we identified nine statistically significant gene expression clusters (Figure 6C) and successfully generated multiple pulse expression profiles. These clusters show the expression patterns of different genes under HT stress. Clusters 1, 3, 4, 5, 6, and 8 presented trends toward downregulated expression, and the cluster with the greatest number of DEGs in the downregulated expression profile was cluster 1, with 5494 genes, which may suggest that this cluster plays an important role in the response to HT stress. The cluster with the fewest DEGs in the downregulated expression profiles was cluster 3, with 480 genes. Although the number of genes involved is small, these genes may also play key roles in specific pathways or processes. Conversely, clusters 2, 7, and 9 tended to show upregulated gene expression under different treatments, and the group with the greatest number of DEGs in the upregulated expression profile was cluster 7, with 2720 genes, which may indicate that these genes were inhibited under HT stress, thus affecting related biological functions. In the upregulated expression profile, the nine classes with the fewest DEGs included 1071 genes. These results revealed the complex response mechanism of *G. barbadense* to HT stress. Changes in different gene expression patterns may regulate the tolerance of *G. barbadense* to HT stress.

### 2.7. Construction and TF Analysis of the Hormone-Related Coexpression Network

To reveal the functions of the regulatory networks, we used a weighted gene coexpression network analysis (WGCNA) of the HT stress-related coexpression networks in *G. barbadense* anthers (Figure 7A,C). The WGCNA results of the DEGs revealed four coexpressed gene modules (Figure 7B). We further calculated the correlations between modules and hormone contents and found that the gene expression profiles in the blue module were highly correlated with the ABA, IAA, and JA levels and that the gene expression profiles in the turquoise module were highly correlated with the IAA, SA, and GA_20_ levels. The blue module was composed of 5 TFs, 5 hub genes of 5 different types (AP2/ERF-AP2, MYB, C3H, C2H2, and bZIP), and the top 100 genes with the highest connectivity, among which the top hub gene was Gbar_A09G027130 (Figure 7D). The turquoise module was composed of 5 TFs, 5 hub genes of 5 different types (AP2/ERF-AP2, NAC, HB-HD-ZIP, and bHLH), and the top 100 genes with the highest connectivity, among which the top hub gene was Gbar_A04G010820 (Figure 7E). To further investigate the transcription factor (TF) expression patterns, we used RNA-seq to analyze the expression patterns of eight TF genes (Appendix A). Our findings identified potential candidate genes for further exploration of the genes related to anther dehiscence in *G. barbadense*.

### 2.8. Influence of HT Stress on TF Expression and Construction of a Genome-Wide Coexpression Network

A total of 1114 TFs were differentially expressed under different treatments, and their expression profiles were clearly distinguished between the upregulated and downregulated groups (Figure 8A–C). We found that the genes of five TF families, including AP2/ERF-AP2, MYB, HB-HD-ZIP, bHLH, and NAC, presented upregulated expression. Conversely, the expression levels of five TF family genes, AP2/ERF-AP2, C2H2, MYB, bZIP, and C3H, were decreased. Because TFs can mediate the inhibition of target gene expression, there should be a high correlation between their expression and the expression profiles of the target genes. We identified correlations between four modules previously detected by WGCNA and various processes. We selected the blue and turquoise modules, which were significantly and highly correlated with different treatments (Figure 8D), and constructed a whole-genome coexpression network of *G. barbadense* using the genes with the top-ranked TFs and connectivity. A total of eight classes of TFs and the top 100 genes with the highest connectivity in each module were identified (Figure 8E). In conclusion, not only were the specific expression patterns of TFs in *G. barbadense* under HT stress revealed, but a genome-wide coexpression network based on hormones, TFs, and key node genes was also successfully constructed. These findings provide a basis for further studies on the molecular mechanism of HT resistance in *G. barbadense*.

### 2.9. Expression Validation and Genetic Variation Analysis of Candidate Genes

Gene expression patterns are usually closely related to gene function. To better understand the expression patterns of the candidate genes, we conducted qRT-PCR experiments on 10 candidate genes in anthers (Figure 9). The results of the qRT-PCR analysis of 10 candidate genes were consistent with the RNA-seq results. The expression levels of Gbar_A09G027130 (ALA4), Gbar_D04G003870 (DUS3), and Gbar_D13G025850 (PI4KG1) significantly decreased under the different treatments.

To further study the 10 candidate genes associated with male sterility in *G. barbadense*, we used the resequencing data of 221 *G. barbadense* materials to type these candidate genes (Figure 10). The results revealed that 9 of the 10 candidate genes presented single-nucleotide polymorphisms (SNPs) in their upstream, downstream, exon, and intron regions. Notably, the Gbar_A07G004670 (APS1) gene had no SNPs at any location, whereas the Gbar_A02G001830 (SBP1) gene had the highest number of SNPS (up to 28). Mutations were identified at positions 21444092 and 21448044 of this gene that resulted in the stop gain of the amino acid sequence. These findings strongly suggest that the Gbar_A02G001830 (SBP1) gene may play a crucial role in the male sterility of *G. barbadense*.

## 3. Discussion

The appropriate temperature is crucial for the entire growth period of cotton [22]. Once the ambient temperature exceeds the threshold that cotton growth can withstand, the plant will be destroyed by heat damage. From the perspective of the plant phenotype, HT can lead to the excessive elongation of plant stems and leaves, increase the leaf area, and affect many apparent traits, such as the fresh and dry weights of leaves, the length and width of the main stem, and the plant height [23]. Moreover, HT stress can also cause the accumulation of reactive oxygen species inside plants, which leads to a significant increase in the level of intracellular oxidative stress [24]. When the temperature is too high, it may even trigger lipid peroxidation of the cell membrane, resulting in rupture and damage to the cell membrane and severely affecting the normal physiological functions of the plant [25]. In response to this oxidative damage, plants activate a series of protective mechanisms, including the activation of antioxidants and the induction of antioxidant enzyme activity [26], to maintain the stability of cell membranes and ensure that plants can continue to grow and develop in HT environments. In this study, among the four flower traits investigated (style length, stigma length, stamen length, and single anther length), only the style length was significantly correlated with stigma length, whereas no significant relationships were found among the other traits. These findings suggest that these flower traits may involve relatively independent regulatory mechanisms during growth and development. Under HT stress, only the stigma length increased, whereas the style length, stamen length, and individual anther length decreased. These results indicate that HT stress has a selective effect on the development of reproductive organs in *G. barbadense*, in which stigma length may have a special adaptive effect, whereas other floral traits are sensitive to HT. The reproductive capacity or adaptability of *G. barbadense* may be enhanced by altering the development of several flower traits (such as increasing stigma length) under HT stress. The tZR, IAA, and GA_20_ levels, which are closely related to plant growth and development, are significantly correlated with many indices of flower traits. These findings suggest that these metabolites may play important roles in flower formation and development, especially in the regulation of style and stamen length. The levels of three hormones, ABA, JA, and SA, which are related to the plant stress response and defense response, were not significantly correlated with flower traits. This does not mean that these hormones have no role in plant growth and development or may not directly participate in or mainly affect the development of flower traits; rather, they may participate more in the plant stress response under adverse conditions.

Male sterility, especially abnormal anther dehiscence, is usually the result of multiple metabolic pathways regulated by the expression of multiple genes (including TFs) [27]. Among these complex regulatory mechanisms, plant hormones play crucial roles, especially ABA, which plays an irreplaceable role in the development of plant reproductive organs [28]. The accumulation of ABA in cotton often leads to male sterility [29]. In tomato, high concentrations of ABA may also lead to abnormal anther development [30]. The ABA content increased significantly after HT stress. By screening differentially expressed genes (DEGs), we identified 14 key pathway genes in eight classes. Most of the genes (nine) tended to have downregulated expression, whereas a few genes (five) tended to have upregulated expression. The results also revealed that in response to HT stress, the accumulation of ABA was negatively regulated by the expression of ABA response element binding factor (ABF) and violaxanthin decycloxygenase (VDE). Moreover, the expression of CYP707A (ABA 8′-hydroxylase) and NCED (9-cis-epoxide carotenoid dioxygenase) was induced to positively regulate ABA accumulation.

Jasmonic acid (JA) is an essential plant hormone regulator that plays a crucial role in the HT stress response [31]. In the biosynthetic pathway of JA, genes such as those encoding acyl-CoA oxidase (ACO) and propylene oxide cyclase (AOC) are considered of special importance because they are involved in the production of sterile male plants during HT stress [32]. Histone H3K27me3 also affects the male fertility of cotton by inhibiting JA synthesis and signaling [33]. Aamir Hamid Khan et al. [34] reported that a decrease in the JA content resulted in abnormal microspore development, pollen inactivation, and anther nondehiscence. In this study, the JA content significantly increased after HT stress. However, the expression levels of most of the pathway genes involved in JA biosynthesis and signaling were decreased (11 genes, with 8 genes downregulated). These results indicate that most of the genes that may regulate JA content are negative regulatory genes that play important roles under stress conditions.

Auxin, a plant growth hormone, plays crucial roles in the growth and development of plants and in coping with stress conditions [35]. Two TFs, ARF6 and ARF8, regulate the expression patterns of auxin response genes by binding to their promoter elements. However, in double loss-of-function mutants of ARF6 and ARF8, plants exhibit flower stagnation, stamen shortening, and anther nondehiscence and are thus unable to release pollen grains [36]. In our study, we observed a significant decrease in the IAA content in the plants after HT stress. Further analysis revealed that most of the genes related to IAA biosynthesis and signal transduction presented downregulated expression (49 genes involved, 35 of which presented significantly downregulated expression), and the plants in the present study also presented male sterility and no anther dehiscence. However, in contrast to previous studies, the changes in the IAA content under HT stress observed in this study were not consistent. We speculate that this may be because the interaction and regulatory mechanisms between different hormones in plants responding to HT stress may be more complex and need to be further explored.

The role of cytokinin is often neglected in studies of the mechanism of male sterility caused by HT stress. However, in this study, we observed that tZR (zeosin riboside, a major cytokinin) levels were significantly downregulated under different treatments, and the expression levels of the genes related to the intrinsic regulation of tZR also significantly differed. These findings suggest that tZR may play an important role in the phenomenon of male sterility under HT stress. Therefore, the effect of tZR on male sterility under HT stress is worthy of further study and confirmation.

GA20, an endogenous regulator [37], has significant effects on multiple developmental processes in plants, including the development of seeds, anthers, and pollen and the response of plants to various abiotic stresses [38,39]. In a previous study, researchers reported that a lack of GA (gibberellin) can significantly affect the normal development of stamens and that the overactivation of GA signals can also adversely affect the fertility of plants [40]. Under high-temperature stress, the DELLA protein encoded by the core gene of the germ plays a key regulatory role [41], which effectively increases the survival rate of plants under abiotic stress by regulating the scavenging activity of reactive oxygen species (ROS). We found that the content of GA_20_ (a key intermediate in the gibberellin synthesis pathway) was significantly decreased after HT stress. Further analysis revealed that the core regulatory genes DELLA and GA2OX encoding GA_20_ were also significantly downregulated, which usually implies the inhibition of gibberellin synthesis and signal transduction pathways. However, simultaneously, the expression levels of GID (gibberellin-insensitive dwarfing protein) and PIF (photochromic interaction factor) were significantly increased. The synergistic effect of the expression of these genes led to a decrease in the GA_20_ content, which affected the male fertility of plants and produced male-sterile plants.

Under high-temperature stress, various hormones such as ABA, JA, IAA, tZR, and GA_20_ exhibit complex interactions and synergistic regulatory mechanisms in *G. barbadense* [42]. These mechanisms collectively influence anther dehiscence, pollen viability, and male fertility. As a stress-responsive hormone, ABA accumulates under high temperatures and works synergistically with other hormones like JA to regulate the plant′s stress response. Meanwhile, the coordinated changes in IAA, GA_20_, and tZR directly affect anther dehiscence and pollen release, with their reduced levels potentially being a key factor in male sterility. Under high-temperature stress, these hormonal interactions and synergistic regulations are disrupted, further exacerbating male sterility in *G. barbadense*.

WGCNA is an effective technique for classifying transcriptome data into coexpression modules to reduce the number of potential candidate genes [43,44,45,46]. In this study, we identified four modules, among which the blue and turquoise modules were strongly correlated with the male sterility of *G. barbadense*. Through transcriptomic and hormone profile analysis of *G. barbadense* tissue, we established a coexpression network of hormones, TFs, and their target genes under HT stress. The synergistic relationship between hormone signals under HT stress was revealed, providing valuable insights into the molecular mechanism of male sterility in *G. barbadense*.

By integrating transcriptomic and metabolomic data, researchers can gain a deeper understanding of biological processes such as plant growth, development, and stress responses [47]. This comprehensive analysis not only helps to reveal the regulatory networks between genes and metabolism but also further explores the complex biological mechanisms that occur within plants. In this experiment, we selected two modules (blue and turquoise) that were significantly correlated with hormones at different times and identified the top five genes in each module as candidate genes (Appendix A). Among these genes, the ALA4 (Gbar_A09G027130) gene (ranked first in the blue module) has been reported to be associated with male fertility in *Arabidopsis thaliana* [48]. Two mutations in the Gbar_A02G001830 (SBP1) gene lead to the acquisition of stop codons in the amino acid sequence, and these two mutations can be used as candidate gene markers.

To gain a more comprehensive understanding of the molecular mechanisms underlying male sterility in *G. barbadense* under high-temperature stress, future research should delve deeper into the interactions [49] and synergistic regulatory mechanisms [50] among these hormones. On one hand, focusing on candidate genes closely associated with male sterility (such as ALA4 and SBP1), functional studies using gene editing and expression pattern analysis [51] could elucidate their specific roles under high-temperature stress, revealing how they regulate anther dehiscence and pollen viability. On the other hand, multi-omics approaches, including transcriptomics and metabolomics, should be employed to construct and refine hormone regulatory network models [52]. This would help analyze the interactions and coordinated regulation of key hormones (such as ABA, JA, IAA, tZR, and GA20) under high-temperature stress, clarifying how these hormones influence male fertility in *G. barbadense* through complex signal transduction pathways [53].

This study not only summarizes the interaction network of hormones, transcription factors, and genes related to HT stress in *G. barbadense* but also provides a basis for analyzing the molecular mechanism of male sterility in *G. barbadense*.

## 4. Materials and Methods

### 4.1. Plant Materials and Growth Conditions

In this study, *G. barbadense* GB150 was used as the test material (this material is an HT-sensitive material, and the anther does not dehisce following HT stress). The test was conducted in 2023 in Awati County (80°39’ E, 40°39’ N). A warming shed was built to stress GB150 in the field, and GB150 materials in the natural state of the field were used as normal controls. In accordance with the weather conditions in Awati County and the growth of *G. barbadense*, field warming treatment was performed for 10 days during the cotton blooming period (9 July 2023–18 July 2023). The temperature was increased every day from 11:00 to 20:00. Thermometers were hung 30 cm above the canopy inside and outside the greenhouses. The average temperature increase during the stress period was 3.645 °C (Appendix A).

### 4.2. Determination of Flower Organ Indices

The stylar length, stigma length, stamen length, and single anther length were measured after HT stress.

### 4.3. Determination of Plant Hormone Contents

Anther samples of *G. barbadense* GB150 stored at ultralow temperature were removed and ground (30 Hz, 1 min) with a grinder until powdered. Fifty milligrams of ground sample was weighed, 10 μL of internal standard mixed solution with a concentration of 100 ng/mL, and 1 mL of methanol/water/formic acid (15:4:1, v/v/v) extractant were added, and the mixture was mixed well. The mixture was vortexed for 10 min and then centrifuged at 4 °C and 12,000 r/min for 5 min, after which the supernatant was transferred to a new centrifuge tube for concentration. After concentration, the mixture was redissolved in 100 μL of 80% methanol/aqueous solution, and the concentration was filtered through a 0.22 μm filter. The contents of ABA, JA, SA, tZR, IAA, and GA_20_ were determined by liquid chromatography–tandem mass spectrometry (LC-MS/MS) analysis.

### 4.4. Transcriptomic Sequencing

Three biological replicates were used to construct RNA-seq libraries of 6 samples from GB150 anther tissue of *G. barbadense* under HT stress compared with a normal control, and sequencing was performed using an Illumina platform. Strict quality control was performed using fastp, and high-quality data were used for subsequent analysis. The *G. barbadense* 3-79 genome (https://cottonfgd.net (*G. barbadense*, HAU)) (accessed on 10 December 2024) [54] was used as a reference genome. Clean reads were sequentially compared with the reference genome in HISAT2 to obtain their position information on the reference genome or gene. Finally, the reads were assembled into transcripts using StringTie.

### 4.5. Coexpression Network Construction

We used the pickSoftThreshold function in the WGCNA package in R to calculate the optimal power value (the RsquaredCut value was set to 0.85), which identified 4 different expression modules. Finally, Cytoscape (v3.10.0) software was used to visualize the coexpression network.

### 4.6. qRT-PCR

Primer Premier 5 software was used for primer design, and primer specificity was detected using DNAMAN (9.0) software. qRT-PCR analysis was performed using an Applied Biosystems 7500 Fast real-time quantitative PCR instrument (Applied Biosystems, Foster City, CA, USA) and a Thermo SYBR qPCR Master Mix kit (Thermo Fisher Scientific (China) Co., Ltd., Shanghai, China) according to the instructions. Each template was set up for 3 biological replicates, and the relative gene expression was calculated using the 2^−∆∆Ct^ method.

## 5. Conclusions

Through integrated transcriptomic and metabolomic analysis, this study investigated the molecular mechanisms underlying high-temperature-induced male sterility in *Gossypium barbadens*. Our findings demonstrate that elevated temperatures significantly altered the phytohormone profiles, with marked increases in the ABA and JA levels accompanied by significant decreases in the IAA, tZR, and GA20 concentrations, while the SA content remained stable. Our comprehensive analysis identified multiple differentially expressed genes associated with hormone biosynthesis and signaling pathways. Utilizing weighted gene coexpression network analysis (WGCNA), we established hormone interaction networks and genome-wide coexpression patterns, revealing 10 candidate genes potentially involved in male sterility. The subsequent validation through qRT-PCR and polymorphism analysis highlighted the functional significance of the ALA4 and SBP1 genes in the sterility phenotype. These results provide novel insights into the molecular basis of thermosensitive male sterility in *Gossypium barbadens* and offer valuable genetic resources for developing heat-tolerant *Gossypium barbadens* cultivars through molecular breeding approaches.

## Figures and Tables

**Figure 1 ijms-26-03693-f001:**
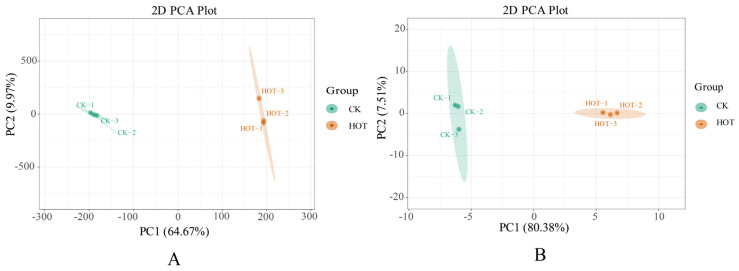
Principal component analyses of transcriptomic and metabolomic samples. (**A**) PCA of transcriptomic samples. (**B**) PCA of metabolomic samples. HT: High-temperature, CK: control check, the same as below.

**Figure 2 ijms-26-03693-f002:**
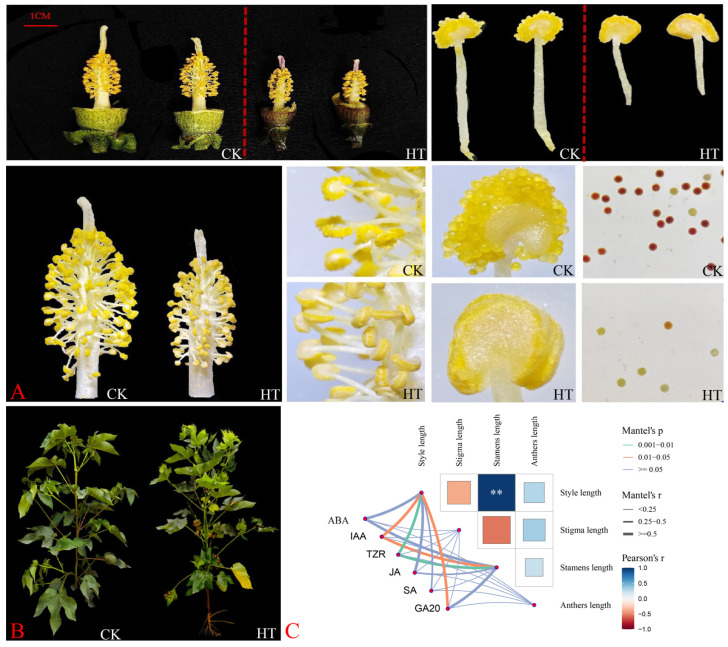
Response of GB150 to HT stress. (**A**) Flower phenotypes of the GB150 germplasm after HT treatment. (**B**) Phenotypes GB150 germplasm HT-treated plants. (**C**) Correlation network of 4 floral organ indices and the levels of 6 major hormones in the GB150 germplasm. ABA: Abscisic Acid, JA: Jasmonic Acid, SA: Salicylic Acid, GA_20_: Gibberellic Acid 20, tZR: trans-Zeatin Riboside, IAA: 3-Indoleacetic acid, the same as below ** *p* < 0.01.

**Figure 3 ijms-26-03693-f003:**
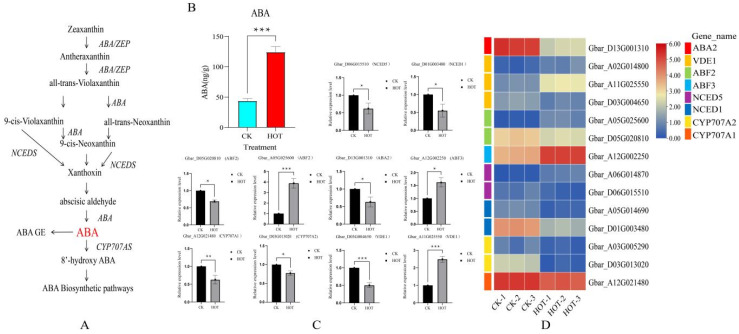
Effects of HT stress on ABA biosynthesis and signal transduction. (**A**) ABA biosynthesis pathway. (**B**) Changes in ABA content under HT stress. (**C**) Expression patterns of 10 ABA pathway genes. (**D**) Expression profiles of genes involved in ABA biosynthesis and signal transduction after HT stress. * *p* < 0.05; ** *p* < 0.01; *** *p* < 0.001.

**Figure 4 ijms-26-03693-f004:**
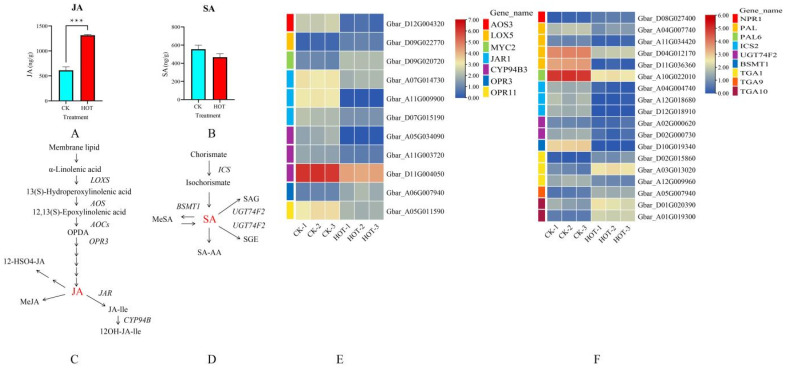
Effects of HT stress on JA and SA biosynthesis and signal transduction. (**A**) Changes in JA content under HT stress. (**B**) Changes in SA content under HT stress. (**C**) JA biosynthesis pathway. (**D**) SA biosynthesis pathway. (**E**) Expression profiles of JA biosynthesis- and signal transduction-related genes after HT stress. (**F**) Expression profiles of genes involved in SA biosynthesis and signal transduction after HT stress.*** *p* < 0.001.

**Figure 5 ijms-26-03693-f005:**
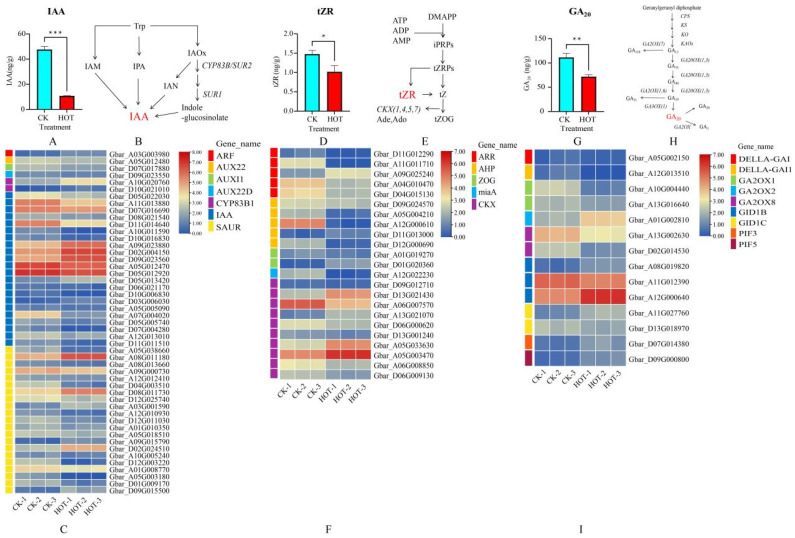
Effects of HT stress on the biosynthesis and signal transduction of IAA, GA_20_, and tZR. (**A**) Changes in IAA content under HT stress. (**B**) IAA biosynthesis pathway. (**C**) Expression profiles of genes involved in IAA biosynthesis and signal transduction after HT stress. (**D**) Changes in TZR content under HT stress. (**E**) TZR biosynthetic pathway. (**F**) Expression profiles of tZR biosynthesis and signal transduction genes after HT stress. (**G**) Changes in the GA_20_ content under HT stress. (**H**) GA20 biosynthetic pathway. (**I**) Expression profiles of genes involved in GA_20_ biosynthesis and signal transduction after HT stress. * *p* < 0.05; ** *p* < 0.01; *** *p* < 0.001.

**Figure 6 ijms-26-03693-f006:**
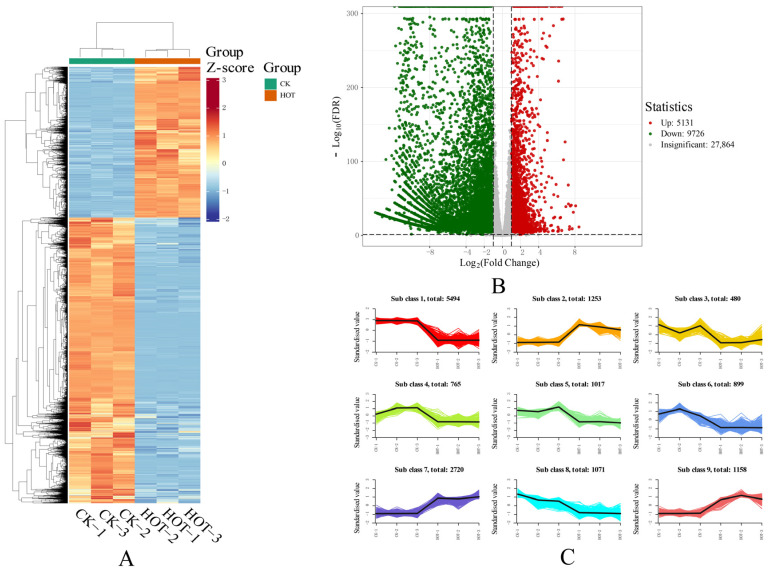
Changes in the whole transcriptome under HT stress. (**A**) Expression patterns of differentially expressed genes under HT stress. (**B**) Volcano map of differentially expressed genes. (**C**) Cluster and line graphs showing the trends of gene expression in the clusters.

**Figure 7 ijms-26-03693-f007:**
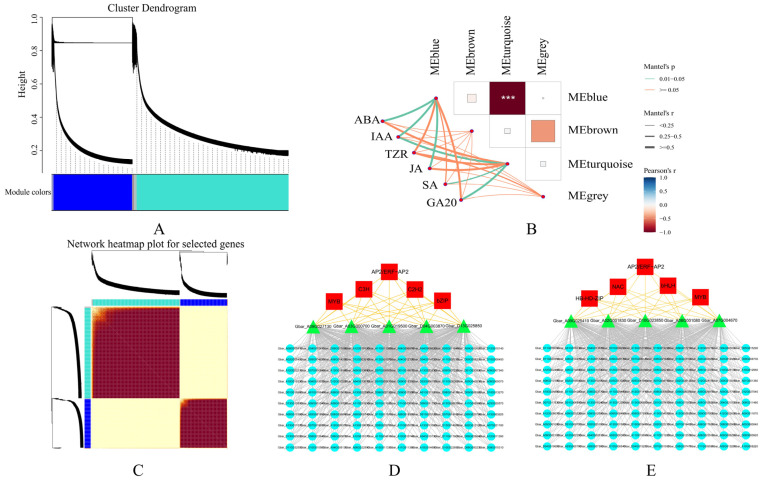
Correlation and coexpression networks between different gene modules and hormones. (**A**) Division of different gene modules. (**B**) Correlations between hormones and modules. (**C**) Modular gene clustering heatmap. (**D**,**E**) Gene coexpression network. *** *p* < 0.001.

**Figure 8 ijms-26-03693-f008:**
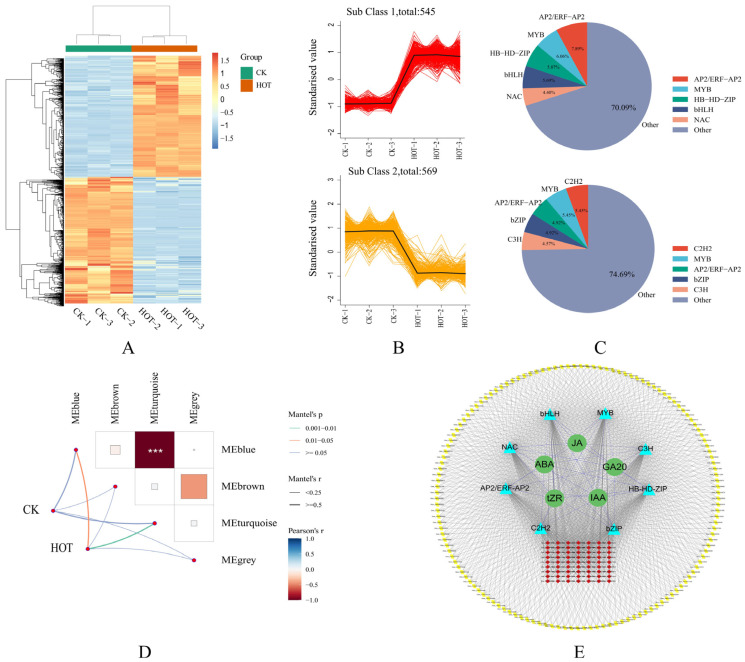
Expression patterns of differentially expressed TFs under HT stress, the correlations between modules and stages, and the whole-genome coexpression network. (**A**,**B**) Heatmaps and line maps of differentially expressed TFs. (**C**) Ratios of TFs whose expression levels were increased to those whose expression levels were decreased. (**D**) Heatmap of the relationships between modules and time points. (**E**) Genome-wide coexpression network. *** *p* < 0.001.

**Figure 9 ijms-26-03693-f009:**
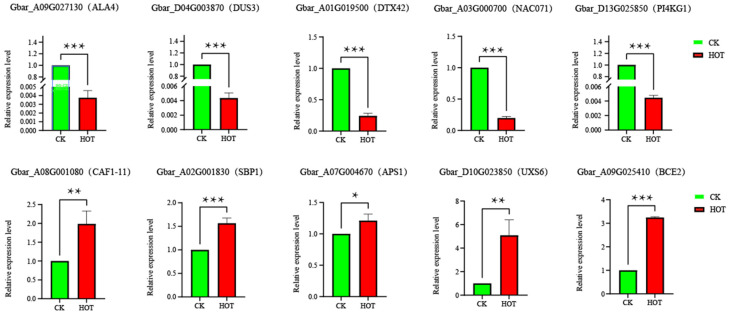
Analysis of the specific expression profiles of 10 candidate genes in anther tissues. * *p* < 0.05; ** *p* < 0.01; *** *p* < 0.001.

**Figure 10 ijms-26-03693-f010:**
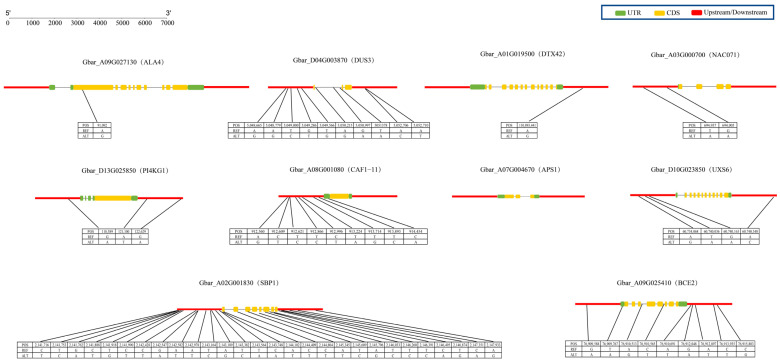
Information on SNP variations in 10 candidate genes in 221 *G. barbadense* accessions.

**Table 1 ijms-26-03693-t001:** Statistical results of the sequencing data.

Sample	Raw Reads	Reads Mapped	Clean Reads	Clean Base (G)	Error Rate (%)	Q30 (%)	GC Content (%)
CK-1	47,820,474	40,368,138 (94.21%)	42,847,734	6.43	0.01	95.2	44.46
CK-2	74,758,782	65,763,208 (93.82%)	70,097,786	10.51	0.02	94.28	44.44
CK-3	88,691,976	77,567,651 (94.88%)	81,755,950	12.26	0.01	96.38	44.29
HOT-1	99,523,708	87,674,236 (95.02%)	92,264,390	13.84	0.01	97.19	43.76
HOT-2	87,482,908	77,555,852 (95.87%)	80,895,212	12.13	0.01	97.12	43.86
HOT-3	49,351,338	41,977,255 (95.83%)	43,803,982	6.57	0.01	96.2	44.14

## Data Availability

The original contributions presented in the study are included in the article/Appendix A. Further inquiries can be directed to the corresponding author.

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
