# Peer review of "Multiomics Studies on the Effects of High-Temperature Stress on Male Sterility in Gossypium barbadense"

_ijms, 2025, doi:10.3390/ijms26083693_

Round 1

Reviewer 1 Report

Comments and Suggestions for Authors

The authors of “Multiomics studies on the effects of high-temperature stress on male sterility in Gossypium barbadense” presents a comprehensive multi-omics study on the impact of high-temperature (HT) stress on male sterility in Gossypium barbadense. The research is timely and relevant, given the increasing threat of climate change-induced HT stress to cotton production. The use of transcriptomic and metabolomic analyses provides a holistic view of the molecular mechanisms underlying male sterility, which is a significant strength of the study. The authors employ a series of advanced data analysis methods. The quality assessment of transcriptomic and metabolomic sequencing data validates the reliability of the data. The use of weighted gene co-expression network analysis (WGCNA) to identify candidate genes related to male sterility is a sophisticated approach, which helps to narrow down the potential key genes involved in the regulatory network. However, there are some minor aspects that could be improved to enhance the overall quality of the paper.

  1. Why do you choose to stress the temperature between 11:00 and 20:00 every day?What measures were taken to control other environmental variables such as humidity and light, which could also significantly impact the results? Details on the control settings within the warming sheds would be helpful.
  2. Could the authors specify why G. barbadense GB150 was chosen as the primary material among other varieties? What specific traits or previous findings indicated that GB150 would be particularly suitable for studying heat stress-induced male sterility?
  3. A limited number of samples may result in a lack of representativeness in the analysis. The metabolic pathways and gene expression networks in organisms are highly complex, and a small sample size may lead to insufficient depth of analysis. It is recommended to increase the sample size and conduct repeated tests to enhance the representativeness of the samples in future analyses.
  4. After exposure to high temperatures, the male sterile phenotype is produced. Subsequent observations can be conducted to further analyze and obtain more systematic results through sampling.
  5. GB150 material produced sterility phenotype after high temperature stress, have you continued to observe how long this sterility phenotype phenomenon has lasted
  6. Figure 3D, what do the colours mean, fold induction compared to what? Please explain.

Author Response

Thank you for your meticulous review and constructive criticism. We have carefully considered all your comments and have implemented revisions to address the points raised.

Comments 1:Why do you choose to stress the temperature between 11:00 and 20:00 every day?What measures were taken to control other environmental variables such as humidity and light, which could also significantly impact the results? Details on the control settings within the warming sheds would be helpful.

Response 1: Regarding the layout of the heating shed, we have extensively consulted a large number of literatures to find the most suitable heating method. The following are the details of the more suitable heating shed construction that we summarized and implemented in the field:The materials in the natural state of the field were used as the normal control.  According to the weather conditions in Awati County and the growth of Gossypium barbadense, field warming treatment was carried out for 10 days during the blooming period of Gossypium barbadense (2023.7.9-2023.7.18).  The planting mode of 1 film 6 rows (66+10)cm was adopted in the warming shed plot, the plant distance was 10 cm, the film width was 2.05 m, the length was 2.5m, the warming shed plot was 27.6m in length and 2.5m in width, the plot area was 69㎡, and two replicates were set with a total area of 138㎡.  The temperature increasing shed is supported by 1.8m high wooden piles, and the shed body is covered by Polyethylene film with 0.08mm thickness and 80% light transmittance (there is a gap of 30cm between the shed film and the ground to ensure gas exchange).  Temperature increase stress was carried out between 11:00 and 20:00 (Some studies have reported that nighttime high temperature damage to Gossypium barbadense is greater than daytime high temperature, in order to fit the actual production to reduce the impact of nighttime high temperature on Gossypium barbadense, so there is no all-day high temperature stress).  Thermometers were hung 30cm above the canopy inside and outside the greenhouse.  The average temperature increase during stress was 3.645℃ (Fig.S3).

Comments 2:Could the authors specify why G. barbadense GB150 was chosen as the primary material among other varieties? What specific traits or previous findings indicated that GB150 would be particularly suitable for studying heat stress-induced male sterility?

Response 2: First of all, the heat resistance of the resource materials was identified indoors and in the field, and the joint identification results showed that GB150 was a heat-sensitive material (Fig.S3). Then, we applied high temperature stress to the material in the field by using a warming shed. Compared with other materials, GB150 material showed anther non-cracking and male sterility (Fig.2-2). Therefore, we have carried out a more in-depth study of GB150 materials.

Comments 3:A limited number of samples may result in a lack of representativeness in the analysis. The metabolic pathways and gene expression networks in organisms are highly complex, and a small sample size may lead to insufficient depth of analysis. It is recommended to increase the sample size and conduct repeated tests to enhance the representativeness of the samples in future analyses.

Response 3: We sincerely thank the reviewers for their professional suggestions regarding sample representativeness.  We have fully recognized the complexity of metabolomics and gene expression network analysis, and have increased the number of samples and replicates in subsequent experiments to ensure greater analytical depth and robustness of conclusions based on an expanded sample size.

Comments 4:After exposure to high temperatures, the male sterile phenotype is produced. Subsequent observations can be conducted to further analyze and obtain more systematic results through sampling.

Response 4: We would like to express our gratitude to the reviewers for their valuable suggestions on deepening phenotypic tracking and multi-omics analysis.  We will make improvements to subsequent experiments to ensure the systematic nature of our research conclusions and the depth of mechanistic analysis.

Comments 5:GB150 material produced sterility phenotype after high temperature stress, have you continued to observe how long this sterility phenotype phenomenon has lasted

Response 5: After the male sterile phenotype was generated, we continued to observe it for 10 days. During these 10 days, the sterile phenotype was observed every day.

Comments 6:Figure 3D, what do the colours mean, fold induction compared to what? Please explain.

Response 6: In Figure 3D, different colors represent different gene names, and the same color represents different gene ids of the same gene

We hope that our answers are satisfactory to all teachers, and we will continue to revise them if there is any deficiency. we very much hope that this paper can be published in your journal.Thanks again to the editor and reviewers for your careful guidance and suggestions to make this article even better.

Finally, I wish all teachers good health and smooth work!

Reviewer 2 Report

Comments and Suggestions for Authors

Li et al., provide valuable insights for the effects of high-temperature stress on male sterility in Gossypium barbadense focusing on hormonal activities by using multiomic technology. The manuscript is well written according to journal’s aims and scope; applied experimentation is based on latest standards. The manuscript is title-focused, well-structured, well-written and it covers all potential aspects of male sterility; omics technology was set properly in order to provide coherent series of data which end up to solid conclusions; experimental design covers - in full - all topic aspects.  Scientific references supported the manuscript and explained clearly the presented data. Examination of four flower traits secures the conclusive outcome of the findings; authors took special provisions that the results are independent of occasional hormonal balance in a narrow genetic basis study. The manuscript couples’ hormonal analytics, with potential novel solutions for male sterility in Gossypium barbadense under heat stress. Gene expression data are well integrated with phenotypic results as shown in the results section. Signaling is a very important part to solidify understanding for the problem and forthcoming targeted solutions which are also well documented. No major weaknesses were found on the text and/or supplementary material; however,

  1. The manuscript does not have numbered text lines as journal template requests.
  2. The description of the supplementary material at the end of the manuscript do not follow the rules of the provided by journal template (https://www.mdpi.com/files/word-templates/ijms-template.dot)
  3. Figures have different style text than the rest of the manuscript.
  4. In discussion, “JA:” and “IAA:” paragraph introductory abbreviations could be omitted so that the text runs smoothly for the reader.

Author Response

Thank you for your meticulous review and constructive criticism. We have carefully considered all your comments and have implemented revisions to address the points raised.

Comments 1:The manuscript does not have numbered text lines as journal template requests.

Response 1: Thank you for your feedback.  We have revised the manuscript to include numbered text lines as specified in the journal’s template guidelines.  

Comments 2:The description of the supplementary material at the end of the manuscript do not follow the rules of the provided by journal template (https://www.mdpi.com/files/word-templates/ijms-template.dot)

Response 2: Thank you for bringing this to our attention.  We have carefully reviewed the journal’s template guidelines (https://www.mdpi.com/files/word-templates/ijms-template.dot) and updated the description of the supplementary material at the end of the manuscript to ensure full compliance with the specified formatting rules. 

Comments 3:Figures have different style text than the rest of the manuscript.

Response 3: Thank you for highlighting this issue.     We have thoroughly reviewed all figures and confirm that the text elements (including axes labels, legends, and annotations) are consistently formatted in Times New Roman, which matches the font style used throughout the manuscript. Upon investigation, we observed that the perceived discrepancy may have arisen from:Rendering Differences: Variations in how software (e.g., PDF viewers, image editors) renders embedded fonts, particularly at different zoom levels. File Compression: Minor artifacts introduced during the figure export or compression process, which could affect text clarity.We have uploaded clear original images of all the pictures in the attachment

Comments 4:In discussion, “JA:” and “IAA:” paragraph introductory abbreviations could be omitted so that the text runs smoothly for the reader.

Response 4: Thank you for your helpful suggestion.  We have revised the discussion section by removing the ‘JA:’ and ‘IAA:’ abbreviations at the beginning of the paragraphs to ensure the text flows smoothly and remains reader-friendly.

We hope that our answers are satisfactory to all teachers, and we will continue to revise them if there is any deficiency. we very much hope that this paper can be published in your journal.Thanks again to the editor and reviewers for your careful guidance and suggestions to make this article even better.

Finally, I wish all teachers good health and smooth work!

Reviewer 3 Report

Comments and Suggestions for Authors

The study explores the regulatory mechanisms of male sterility-related hormones in Gossypium barbadense under high-temperature stress by integrating transcriptomic and metabolomic analyses and identifies candidate genes closely related to male sterility, providing a theoretical basis for understanding the molecular mechanisms of male sterility in Gossypium barbadense. After revision, the manuscript can be considered for publication.

  1. The first mention of GB150 in the text (e.g., in the introduction) should include a brief introduction to the source and characteristics of this material.
  2. Since the accuracy of phenotypic traits is crucial, what instruments or methods were used to measure “Stylar length, stigma length, stamen length and single anther length”? Please provide detailed information.
  3. Please add the literature source for the G.barbadense 3-79 genome.
  4. In the “Results and analysis” section, references are generally not cited. Sentences such as “Many studies have shown that HT stress has significant effects on many aspects of plant morphology, photosynthesis and the antioxidant system (Ma et al., 2017)” and “JA and SA, as regulators of HT stress, also play important roles in the response to HT (Feng et al., 2018; Yang et al., 2020)” should be removed. Necessary descriptions should be placed in the “Introduction” or “Discussion” sections.
  5. The title “2.2. Effects of different hormones on the flower traits of G. barbadense” does not quite match the content below it, indicating poor logic in the article. All such issues in the text should be carefully verified and revised.
  6. What material was used as the control for “upregulated or downregulated expression”? Please clarify.
  7. In section “2.3.,” how were the 8 core genes selected for qRT‒PCR?
  8. The first paragraph of the “Discussion” should be deleted and revised to be placed in the introduction or results sections. The discussion can be further expanded to delve into key issues. For example, the interactions between hormones and their synergistic regulatory mechanisms in male sterility could be analyzed in greater depth by integrating more literature and theory. Additionally, specific future research directions based on the findings of this study could be highlighted, such as in-depth functional studies of specific genes or further dissection of hormone regulatory networks.
